# Genetic Variation in Complex Traits in Transgenic α-Synuclein Strains of *Caenorhabditis elegans*

**DOI:** 10.3390/genes11070778

**Published:** 2020-07-11

**Authors:** Yiru A. Wang, Lisa van Sluijs, Yu Nie, Mark G. Sterken, Simon C. Harvey, Jan E. Kammenga

**Affiliations:** 1Laboratory of Nematology, Wageningen University & Reseach, Droevendaalsesteeg 1, 6708PB Wageningen, NL, The Netherlands; yiru.wang@dzne.de (Y.A.W.); lisa.vansluijs@wur.nl (L.v.S.); mark.sterken@wur.nl (M.G.S.); 2Biomolecular Research Group, Canterbury Christ Church University, North Holmes Road, Canterbury CT1 1QU, UK; yn280@mrc-mbu.cam.ac.uk (Y.N.); simon.harvey@canterbury.ac.uk (S.C.H.)

**Keywords:** genetic background, α-synuclein, protein accumulation, *C. elegans*

## Abstract

Different genetic backgrounds can modify the effect of mutated genes. Human α-synuclein (*SNCA*) gene encodes α-synuclein, and its oligomeric complexes accumulate with age and mediate the disruption of cellular homeostasis, resulting in the neuronal death that is characteristic of Parkinson’s Disease. Polymorphic variants modulate this complex pathologic mechanism. Previously, we constructed five transgenic introgression lines of a *Caenorhabditis elegans* model of α-synuclein using genetic backgrounds that are genetically diverse from the canonical wild-type Bristol N2. A gene expression analysis revealed that the α-synuclein transgene differentially affects genome-wide transcription due to background modifiers. To further investigate how complex traits are affected in these transgenic lines, we measured the α-synuclein transgene expression, the overall accumulation of the fusion protein of α-synuclein and yellow fluorescent protein (YFP), the lysosome-related organelles, and the body size. By using quantitative PCR (qPCR), we demonstrated stable and similar expression levels of the α-synuclein transgene in different genetic backgrounds. Strikingly, we observed that the levels of the a-synuclein:YFP fusion protein vary in different genetic backgrounds by using the COPAS™ biosorter. The quantification of the Nile Red staining assay demonstrates that α-synuclein also affects lysosome-related organelles and body size. Our results show that the same α-synuclein introgression in different *C. elegans* backgrounds can produces differing effects on complex traits due to background modifiers.

## 1. Introduction

The phenotypic effect of mutations can be modified by the genetic background. This means that mutational effects can be enhanced or mitigated by genetic background modifiers. For example, variants in the *FBN1* gene are responsible for most cases of marfan syndrome, but the phenotypic effect of alleles of *FBN1* is modified by other genes such as *gMod-M1-9* and *COL4A1* [1]. Similarly, multiple variants have been identified from genome-wide association studies that are associated with Aβ and ultimately with the progression and pathology of Alzheimer’s disease (AD) [2]. There is therefore a general consensus that the overall effect of disease-causing mutations is often mediated by the genetic background.

Genome wide screens and sequencing based approaches offer an unbiased route to the identification of genetic factors that modify complex disease traits in humans. Despite progress in such direct human studies, there is still an important role to be played by studies in model species like the nematode *C. elegans*, particularly when the relative ease with which their genetics can be modified is considered. Paaby et al. [3] revealed that variation in embryonic lethality was attributable to heritable modifiers among *C. elegans* wild-type strains, half of which were gene-specific modifiers—i.e., cryptic genetic variation (CGV)—affecting developmental processes. Moreover, the polymorphic gene expression regulation underlying CGVs can occur in cis- or trans-acting regulatory mechanisms in response to rapid temperature changes, which was determined by a transcriptional analysis in *C. elegans* recombinant inbred lines [4]. Similarly, another transcriptome analysis of a *C. elegans* recombinant inbred line (RIL) population containing a gain-of-function mutation in gene *let-60* mapped six major regulators specific for the interaction between the genetic background and the mutation.

We recently introgressed the *pkIs2386* transgene, which produces the expression of a human α-synuclein and YFP fusion in body wall muscle, from the isolate NL5901 (N2 genetic background) into five different *C. elegans* genetic backgrounds (αS-ILs) [5]. Prior to this, the analysis of α-synuclein expression and aggregation in *C. elegans* had always been conducted in the Bristol N2 genetic background. With a single genetic background, it was not therefore possible to assess the role that different backgrounds might play in the consequences of transgene expression [6,7]. Compared to the corresponding wild-type strains, the αS-ILs and NL5901 exhibited a wide range of phenotypes, indicated that each genetic background contained modifiers that altered the response to the effects of the α-synuclein and YFP fusion [5]. Transcriptome analyses of these nematodes revealed genotype-specific effects on α-synuclein-associated life-history phenotypic traits [5].

Here, we have further investigated how the expression of this α-synuclein and YFP fusion, and the subsequent effects of this expression, are affected by the genetic background. Specifically, we measured the α-synuclein transgene expression, the overall levels of the α-synuclein:YFP fusion protein, and the levels of lysosome-related organelles in the αS-ILs, in NL5901, and in the relevant control lines. These results indicate that studies on natural variation beyond the Bristol N2 genetic background could facilitate *C. elegans* as a model of α-synuclein for further understanding α-synuclein toxicity.

## 2. Materials and Methods

### 2.1. C. elegans Strain and Maintenance

Next to N2 and NL5901, four wild-type strains—JU1511, JU1926, JU1931, JU1941—and their corresponding ILs carrying α-synuclein introgression (αS-ILs), SCH1511, SCH1926, SCH1931, SCH1941, were used (as in [5]). These worms contain the transgene *pkIs2386* (unc-54p: α-synuclein: YFP + unc-119(+)), and more details have been shown in the Method and Materials of [5].

Standard culturing techniques were used in culturing all the strains for variable assays. The worms were maintained at 20 °C on nematode growth medium (NGM) plates seeded with *Escherichia coli* OP50 [8]. All the assays were initiated using eggs isolated from gravid adults treated with sodium hypochlorite and NaOH [9]. These synchronized eggs were allowed to hatch on NGM plates seeded with *E. coli* overnight at 20 °C—i.e., the time point was defined as t = 0. When those worms reached the expected stage, these assays were then measured or set up with all lines.

### 2.2. RT-qPCR

Two 10 cm NGM plates containing young-adult worms, which were growing at 20 °C for 48 h old after age-synchronizing the population, were washed off the NGM plates by M9 buffer (KH_2_PO_4_, 22 mM; K_2_HPO_4_, 34 mM; NaCl, 86 mM; MgSO_4_, 1 mM). RNA was isolated from the worm pellets by using the Maxwell^®^ 16 LEV simply RNA tissue kit (Promega). An amount of 1 μg of total RNA was reverse transcribed with oligo dT primers using Superscript II reverse transcriptase (Invitrogen), according to the manufacturer’s protocol. A real-time quantitative polymerase chain reaction (qPCR) was performed on a Bio-rad iCycler, using iQ SYBR green with the following primers: α-synuclein forward 5′-ATGGATGTATTCATGAAAGG-3′, α-synuclein reverse 5′-TTCAGGTTCGTAGTCTTGA-3′; eGFP forward 5′-TTTCTGTCAGTGGAGAGGGT-3′, eGFP reverse 5′-CCTGTACATAACCTTCGGGC-3′; one of the reference genes, *Y37E3.8*—forward 5′-ATCCTGGAGGTCGCGGTAAC-3′, reverse 5′-GCGCCAAGATAGGAGCGGAT-3′; another reference gene, *rpl-6*—forward 5′-AGTGCTCCGCTTCTCTGCTT-3′, reverse 5′-AGGTGACTCTGGACCTCGTT-3′. The experiments were conducted in three biological replicates, and each replicate included the technical duplicates of each sample.

### 2.3. Nile Red Assay

In *C. elegans*, intracellular fat droplets can be stained with the fluorescent dye Nile Red. Nile Red stock (500 µg/mL in acetone) was diluted 1:500 in the OP50 solution used as food for each experiment. The freshly diluted Nile Red *E. coli* diet 200 µL was seeded on 6 cm diameter NGM plates. Subsequently, the L1 larvae obtained from bleaching were transferred to the plates and allowed to grow for 2 days (stage L4 larvae). The plates containing Nile Red as well as the feeding worms were protected from the light by wrapping them in aluminum foil.

### 2.4. Worm Sorter

Besides fluorescent imaging, an automated worm sorter, COPAS™ BIOSORT (Union Biometrica Inc., Somerville, MA, USA), was used for automated fluorescence quantification. Briefly, the 2-day- (~48 h) or 3-day- (~72 h) old worms were washed off the plates by the M9 buffer and then were immobilized on ice. To quantify the intensities of the yellow fluorescence signal, the reading parameters used were time-of-flight (TOF) for the *x*-axis and the GREEN/Red peak height for the *y*-axis. The number of measurements per sample was from 100 to 400. Both the 2-day- and 3-day-old worms were analyzed. The 3-day-old populations were a mixture of mainly adults and some embryos. Therefore, the measured embryo data was filtered out via the rate of extinction by the TOF and TOF values (Appendix A).

### 2.5. Statistical Analysis

All the statistical analyses were performed in R (version 3.5.1 64x). For the worm sorter assay, the raw data were read in, processed, and plotted using the COPASutils R package (Shimko and Andersen 2014). Noise in the data—e.g., air bubbles—were filtered out according to the normalization of worm body parameters (Appendix A). The qPCR measurements were transformed and processed for data normalization according to the method described in the study of [9]. Subsequently, pairwise testing was performed using a two-sample independent t-test, not assuming equal variances. Testing over multiple samples was performed by an ANOVA.

### 2.6. Fluorescent Imaging

The formation of aggregates per individual was tracked by imaging fluorescent foci in live animals of SCH1511, SCH1926, SCH1931, SCH1941, and NL5901 using fluorescent microscopy (Olympus IX83 equipped with cellSens Dimension software, Olympus, Southend-on-Sea, UK). The nematodes were age-synchronized and cultured as described above for 2, 3, 4, 5, 6, and 7 days. Around 6 to 8 individuals were mounted on slides and exposed to a 5x dilution of 1M NaN3 (by M9 buffer). Then, images were taken using green (488 nm) channel with 600 ms of expose time. Notably, the worms over 4 days old required less time in NaN3 in imaging. Meanwhile, the correct developmental stages were verified based on the morphological features.

## 3. Results

### 3.1. Stable α-Synuclein Transgene Expression in Different Genetic Backgrounds

The expression levels of α-synuclein in the different genetic backgrounds were assessed by quantitative PCR (qPCR) in 2-day-old worms (after hatching; similar to [5]). Prior to measuring α-synuclein, the stable expression of the reference genes used in the qPCR was confirmed in each genotype (Appendix A). We then characterized the gene expression of the transgene using primers specific to both α-synuclein and YFP. The simultaneous amplification and quantification of both the sequences within the transgene *pkIs2386* showed similar expression levels across the different genetic backgrounds (Figure 1). Therefore, the different genetic backgrounds appear to only have a limited effect on the expression level of the α-synuclein transgene.

### 3.2. Overall Expression of the a-Synuclein: YFP in Different Genetic Backgrounds

We next investigated the overall expression of the α-synuclein:YFP fusion protein. Firstly, the α-synuclein protein was visualized by a fluorescence microscope, mainly focusing on the head and vulva regions of the αS-ILs and NL5901 worms (Figure 2). The increasing abundance of fluorescent α-synuclein protein inclusions with age were present in NL5901 (Figure 2), which, as a characteristic of α-synuclein aggregation, has been documented before [10,11]. This age-related increase in the α-synuclein aggregation was also seen in the four transgenic lines with a wild-type JU strain background (Figure 2). Interestingly, we observed a relatively stronger fluorescent signal in all the αS-carrying worms with wild-isolate JU backgrounds compared to NL5901, especially in young worms (Figure 2). Then, we investigated changes in the levels of the fusion protein with age in the NL5901 and the four αS-ILs by quantifying the overall fluorescence intensity in live 2-day- and 3-day-old worms using a COPAS Biosorter (Union Biometrica) (Figure 3).

Notably, the 2-day-old worms showed significantly different fluorescent signals among the different genetic backgrounds (ANOVA, F = 695.98, *p* << 10^−5^, Appendix A), even though these worms showed similar expression levels of the α-synuclein transgene (Figure 1). Moreover, the older worms clearly presented higher fluorescent signals than the young worms (ANOVA, F = 682.17, *p* << 10^−5^, Appendix A). SCH1941, which is the JU1941-derived αS-IL, displayed the strongest increase, while SCH1931 showed the lowest increase (Figure 3). Meanwhile, in both the 2-day-old worms and the 3-day-old worms, NL5901 exhibited the lowest fluorescence levels over all the strains, as can be seen in the image results (Figure 2). Together, the measurements by Biosorter and the supplemented visualization by microscopy indicates that genetic background impacts the accumulation of α-synuclein, and shows, for 2-day-old worms, that these differences arise via a post-transcriptional mechanism.

### 3.3. α-Synuclein Affects Lysosome-Related Organelles and Body Size

Previous findings showed that α-synuclein-related synaptic pathologies cause neurodegeneration that is involved in lipid membrane interactions [12,13,14]. Alterations in α-synuclein, including the structural binding and lipid specificity, can cause disruptions in the complex network of the synaptic machinery [15,16]. Recently, it was reported that lysosomes are an important regulator of sorting exogenous and endogenous lipids in metabolic processes [17]. Based on that, the effect of α-synuclein expression may affect lipid dynamics, like lysosome in lipid transport and biogenesis. To address this, we visualized and measured lysosome-related organelles in the αS-ILs and the five wild-type strains as well as N2 and NL5901 by Nile Red staining. Nile Red stains vesicular structures (e.g., lysosome-like compartments) in the intestine, which could offer insights into the lysosomal degradation of α-synuclein in *C. elegans*. The fluorescent intensity of Nile Red staining was tracked by the COPAS Biosorter.

The body length, measured as the time of flight (TOF) in the Biosorter, of the worms was assayed in relation to the lysosomal integrity. No differences in body size were found between the 2-day-old NL5901 compared to the N2, whereas a significant difference was observed for the Nile Red signal (Figure 4a,b, Appendix A) (false discovery rate-corrected Tukey test, q-value = 1 and 0, respectively). Subsequently, the 3-day-old NL5901 had a much larger TOF than the N2 (q-value = 0), and its Nile Red signals were also higher than those of the 3-day-old N2 (q-value = 0) (Figure 4a,b, Appendix A). Across the different backgrounds, the αS-ILs exhibited differences in the Nile Red signal compared to the NL5901 (Appendix A) (Figure 4a,b). Moreover, the Nile Red accumulation—i.e., lysosome-related organelles that uptake the dye in *C. elegans* [18,19]—varied among wild-type JU, CB4856, and N2 worms but also were considerately different from those changes in αS-ILs and NL5901 (ANOVA, *p*-value = 0.021). Here, the comparisons of the Nile Red accumulation among our worms indicate that α-synuclein affects lysosome-related organelles differently in the different genetic backgrounds. Hence, regulators associated with lysosomal integrity in variable backgrounds appear to be differently affected by α-synuclein.

## 4. Discussion

Using a *C. elegans* model of α-synuclein (transgene *pkIs2386* (*unc-54*p: α-synuclein: YFP + *unc-119*(+))) expression in different genetic backgrounds [5], we measured the α-synuclein transgene expression, the overall accumulation of the α-synuclein:YFP fusion protein, and the lysosome-related organelle uptake of Nile Red. Our results show that α-synuclein introgression in different *C. elegans* backgrounds modifies these complex traits. In our model, the α-synuclein:YFP fusion protein is under the control of the *unc-54* promoter, and hence the protein is expressed in the body wall muscle [10]. Our results are therefore informative about the effects of the ectopic expression of an aggregation-prone protein.

The observation of the distribution of the α-synuclein-YFP fusion protein indicated that it remains diffusely localized in the NL5901 as well as the αS-ILs ([10], and Figure 2). The accumulation of α-synuclein-YFP into foci has been seen in all the elderly α-synuclein worms due to the intrinsic properties of the α-synuclein protein—e.g., propagation of misfolded α-synuclein [20]. van Ham et al. [10] over time (5, 11, 13, 16 days). The fluorescence measurements in this study on the αS-ILs worms face challenges, including an increase in internal hatching (bagging) from day 5 onward (shown in [5]) and difficulty in clearly visualizing fluorescent foci in the αS-carrying worms. However, our images of the head and vulva region (for day 2, 3, 4, 5, 6) showed the fluorescence is diffused—i.e., a non-aggregating protein—especially in young worms, and also represent the inclusions that α-synuclein forms with age in all adult α-synuclein genotypes. We postulate that in the αS-ILs worms, the number of inclusions and the accumulation of non-aggregating protein with age increases in the head muscles surrounding the pharynx is similar with the spread found in NL5901, but more and premature. This could explain why the pumping rate did not decrease in NL5901 to the extent seen in the αS-ILs—i.e., the N2 background showed less effect [5]. Meanwhile, we anticipate that numerous aggregates appear at the vulva of young-adult αS-ILs and significantly increase in the vulval muscles (Figure 2). In contrast, NL5901 does not show such a strong fluorescent signal in the vulval region (Figure 2). Ultimately, the marked expression of fluorescent α-synuclein fusion could explain the high internal hatching rate in the αS-ILs (presented in [5]).

The strain NL5901 has the lowest fluorescent levels compared to the other strains by COPAS biosorter quantification. Interestingly, the transcription level of the α-synuclein transgenes in NL5901, with the N2 genetic background, was not significantly different compared to the other five genetic backgrounds tested. van Ham et al. [10] identified a set of genes in NL 5901 that acted as suppressors of α-synuclein aggregates inclusion formation, ER/Golgi complex compartments, vesicular transport, and lipid metabolism. They identified that *sir-2.1*, of which the effect was further confirmed in a genetic deletion strain, functions as a NAD-dependent protein deacetylase. Notably, a gene expression profiling analysis of these αS-ILs in different genetic backgrounds with NL5901 and N2 identified another Sir2 paralog, *sir-2.2*. Its expression was highly affected by an interaction between α-synuclein and genotype (PD by genotype interaction, *p* = 0.002) [5]. A novel set of α-synuclein effector genes modulates α-synuclein aggregation and associated effects, including histone demethylase (*spr-5*), lactate dehydrogenase (*ldh-1*), small ribosomal subunit SA protein (*rps-0*), cytoskeletal protein (*act-5*), collapsing response mediator protein (*unc-33*), and choline kinase (*cka-2*) [21]. Among them, the expression of gene *ldh-1* was also found as responding to α-synuclein in the four αS-ILs backgrounds [5]. Although studies have suggested that these modifiers are involved in several key signaling mechanisms [5], the exact modification/regulation of folding and oligomerization leading to α-synuclein aggregates is still not known clearly.

Proteins with variable abundance are likely to be associated with the biological function of polymorphic variants elsewhere on the genome [22]. Our findings match the previous transcription analysis of four αS-ILs and NL5901 [5], as expression changes are detected to be enriched for genes related to regulating protein coding and synthesis. Hence, these transgenic α-synuclein encoding *C. elegans* could provide a more reliable platform to elucidate the molecular interactions involved in α-synuclein aggregation by using different natural genetic backgrounds.

We expect that the inclusions of α-synuclein-YFP fusion, which appear to be unevenly distributed fluorescent spots in *C. elegans* (Figure 2; [10]), could represent the primary cytotoxic α-synuclein species present in post-mortem human Parkinson’s Disease (PD) brain tissue [23]. Several α-synuclein aggregate structures might be included in the process of clump formation and accumulation: α-Synuclein monomers were suggested to play a role in the complex molecular mechanism, leading to cellular dysfunction or the disruption of other molecular or signaling pathways (see the review of [24]); oligomeric and phosphorylated α-synuclein have mainly been viewed as toxic, and the presence of these forms is considered essential for PD [25].

Lipid interaction and membrane-induced helix formation were described as the roles of the extreme *N*-terminus of α-synuclein (reviewed by [26]). In addition to the *N*-terminal region, two more regions, the non-amyloid-β component and C-terminus, of α-synuclein participate in the membrane-binding process [27]. Unfolded monomeric α-synuclein binds to small unilamellar vesicles composed of negatively charged lipids [28]. As vesicles containing negatively charged lipids are impermeable to protein adsorption in general; the further binding of more toxic oligomeric α-synuclein induces membrane permeabilization leading to membrane defects, which could result in more vulnerability to oligomeric α-synuclein binding [29]. Nile Red intensity is correlated with lysosome abundance; we suggest that the lysosome-related organelles’ uptake of Nile Red—i.e., the site of concentration of the dye—implied the susceptibility of worms to the cytotoxicity of a-synuclein expression.

## 5. Conclusions

In this study, genetic divergence in response to the α-synuclein transgene contributes to finding the genotype-phenotype relationship related to α-synuclein-induced traits or toxicity. These phenotypic changes are variable among different genetic backgrounds and they were more pronounced in the wild isolate JU backgrounds than in the N2 background. This suggests that natural genetic variation appears to affect phenotypic traits in the α-synuclein aggregation processing. Further research aiming to unravel these polymorphic loci would include the backcrossing of the different α-synuclein backgrounds with wild-type Bristol N2 and the subsequent generation of RILs in combination with quantitative trait locus (QTL) mapping.

## Figures and Tables

**Figure 1 genes-11-00778-f001:**
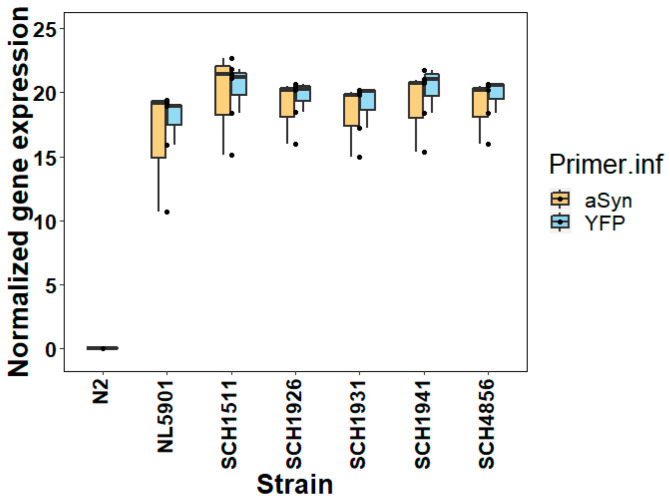
Gene expression levels of α-synuclein transgene in five different genetic backgrounds by qPCR. The relative expression of α-synuclein and fluorescent protein measured by RT-qPCR for these 2-day-old worms is shown; in total, there are three independent repeats in duplicate per experiment. No substantial difference was found between the expression of α-synuclein and YFP for each strain (two-tailed t-test, *p* > 0.05), and no difference between α-synuclein and YFP was found between the strains (two-tailed ANOVA, *p* > 0.05; F value = 1.72 with df = 1 for Primer.inf, and F value = 1.04 with df = 5 for strain).

**Figure 2 genes-11-00778-f002:**
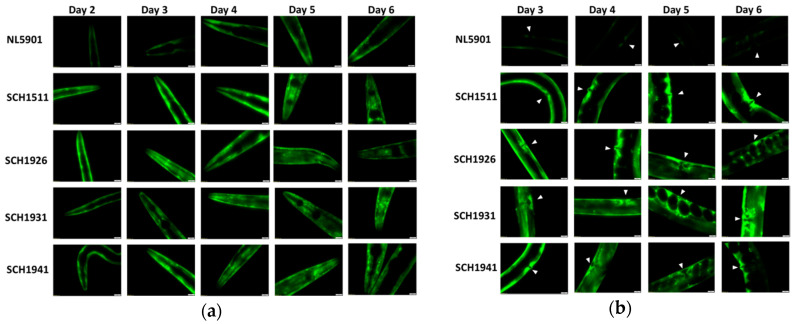
Fluorescent tracking focusing on the head area and vulva area over time among the all αS-carrying worms with wild-isolate JU backgrounds and NL5901. From left to right are different ages of hermaphrodites worms. Exposure time of all the imaging was 600 ms. (**a**) Images show the head region of α-synuclein-YFP transgenic animals. (**b**) Images show the body region of all the hermaphrodites, and arrows denote the position of the vulva.

**Figure 3 genes-11-00778-f003:**
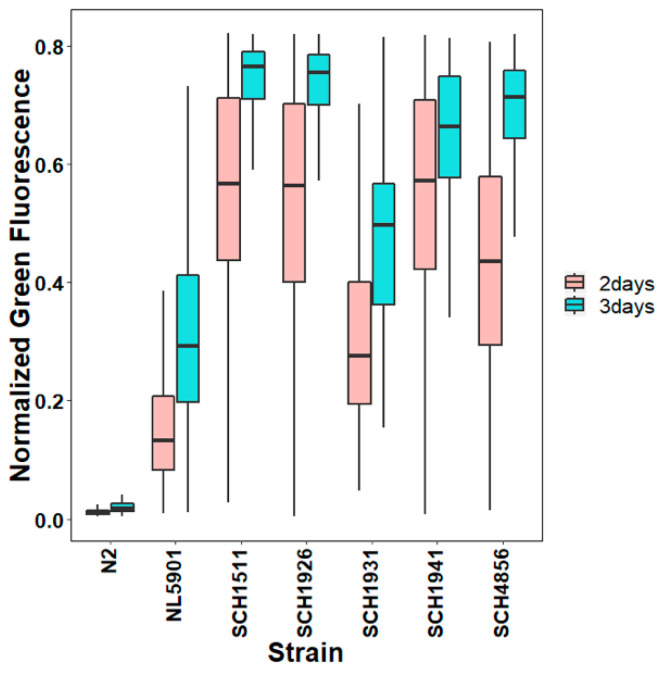
Quantifications of α-synuclein-YFP fusion protein by the COPAS Biosorter, with N2 as the negative control. The six transgenic strains contain α-synuclein but differed among the genetic backgrounds. The fluorescent signal levels were substantially different among 2-day-old worms (ANOVA, *p* << 10^−5^, F value = 695.98 with df = 5), while similarity was among the 3-day-old transgenic worms (ANOVA, *p* << 10^−5^, F value = 682.17 with df = 5). Post-hoc Tukey tests can be found in Appendix A.

**Figure 4 genes-11-00778-f004:**
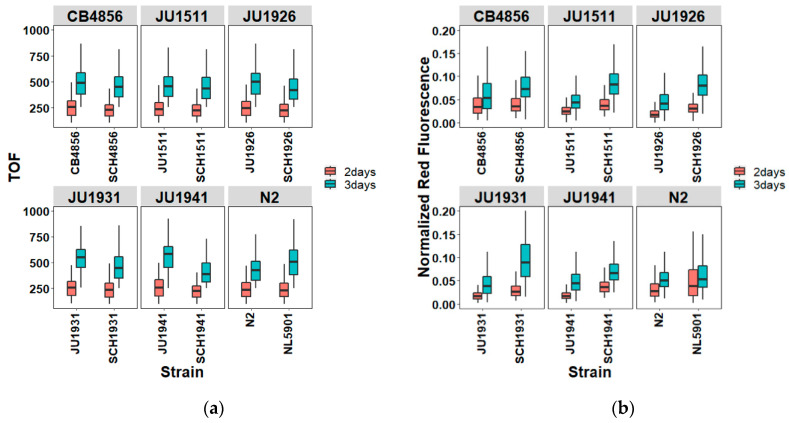
Variation in the body length and Nile Red staining between wild-type worms and their corresponding aS-ILs worms. Both 2-day- and 3-day-old worms were scanned and sorted by the COPAS Biosorter. In total, six different genetic backgrounds were included. (**a**) Measurements of individual body length (time of flight, TOF). For 2-day-old worms, there was no difference showed depending on genotypes (two-way ANOVA, *p* = 0.0696, F value = 2.042 with df = 5), but a difference presented between worms with and without α-synuclein (two-way ANOVA, *p* << 10^−11^, F value = 45.498 with df = 1). The TOF of 3-day-old worms differed by either genotype or expressing α-synuclein (two-way ANOVA, *p* << 10^−11^, F value = 11.577, with df = 5, and *p* <<10^−13^, F value = 52.529, with df = 1). (**b**) Measurements of individual Nile Red staining (Normalized Red Fluorescence). A considerably different fluorescent red signal was affected by genotype in either 2-day- or 3-day-old worms (two-way ANOVA, *p* << 10^−16^, F value = 122.60, and *p* << 10^−16^, F value = 118.49, respectively, both with df = 5) as well as by α-synuclein (ANOVA, for 2-day-old worms, *p* << 10^−16^, F value = 392.69, and for 3-day-old worms, *p* << 10^−16^, F value = 701.40; both with df = 1). The variation in the TOF/Nile Red staining exhibits among worms of N2, JUs, NL5901 and αSILs, of which details can be found in the post hoc Tukey tests (Appendix A). For each treatment and strain, approximately 100–300 worms were studied.

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
