# Peer review of "Genetic Variation in Complex Traits in Transgenic α-Synuclein Strains of Caenorhabditis elegans"

_genes, 2020, doi:10.3390/genes11070778_

Round 1

Reviewer 1 Report

The authors present work to study how genetic background affects several phenotypes driven by expression of α-synuclein. They introgress a transgene driven by an unc-54 promoter into several RILs and study the observed differences in animal size, nile red staining, and transgene expression. The authors conclude genetic background plays a role in the measured parameters, and thus there are interactions between genetic variation and the effects of α-synuclein. The article is well described, and the figures are well presented. The statistical analyses are rigorous and support their findings. My major concerns are:

  1. Figures 1 and 2 relate to expression, which is really studying how genetic background affects expression of unc-54. While the results are surely correct, it is unclear how this relates to alpha-synuclein.
  2. The rationale behind measuring nile red staining in an alpha-synuclein study is not clear. Although much of the discussion is devoted to it, formulating the significance of this specific readout in the results section would help the manuscript, in my opinion.
  3. How are the authors going to use the genetic variants information to reveal the interacting genes?
  4. The discussion is heavily directed to other studies and the supplemental information, rather than the findings of this work on the main text. It is a bit difficult to dissect the significance of these findings. I would suggest emphasizing the findings of this study first, and potential bringing the supplemental results of Figure S3 to the main text.

Reviewer 2 Report

This study asks how different genetic modifiers may change the effects of overexpression of α-synuclein by examining the transgene expression, accumulation of the α-synuclein:YFP, lysosome-related organelles and body size. This study is a nice extension of a previous study Wang et al., 2019, and is of broad interest. The experiments were well designed, and the results were clearly presented.

I have only two related comments that I hope the authors to address:

(1). The rationale to examine the lysosome-related organelles is not very clear to me. In Results 3.3, it says that “Alterations in α-synuclein, including structural binding and lipid specificity, can cause disruptions of the complex network in the synaptic machinery [19,20]. Based on that, the effect of α-synuclein expression may affect lipid dynamics, like lysosome in lipid transport and biogenesis”. First, the connection between lysosomes and lysosome related organelles are not explained here. Second, why do we specifically look at lysosome-related organelles, but not other organelles that are also involved in lipid dynamics? Is it biologically more meaningful? Or is it just because the Nile red assay is easier to do?

(2). I feel that the discussion related to this part (last paragraph) does not help us understand why Nile Red fluorescence increases with the expression of α-synuclein. Is Nile Red fluorescence correlated with the amount of lysosome related organelles? What does this mean in terms of cytotoxicity of α-synuclein expression?
